# Predicted Secretome of the Monogenean Parasite *Rhabdosynochus viridisi*: Hypothetical Molecular Mechanisms for Host-Parasite Interactions

Marian Mirabent-Casals [1], Víctor Hugo Caña-Bozada [1], Francisco Neptalí Morales-Serna [2] and Alejandra García-Gasca [1,*]

[1]  Centro de Investigación en Alimentación y Desarrollo, Avenida Sábalo Cerritos s/n, Mazatlán 82112, Sinaloa, Mexico
[2]  Instituto de Ciencias del Mar y Limnología, Universidad Nacional Autónoma de México, Av. Joel Montes Camarena s/n, Mazatlán 82040, Sinaloa, Mexico
*   Correspondence: alegar@ciad.mx

**Abstract:** Helminth parasites secrete several types of biomolecules to ensure their entry and survival in their hosts. The proteins secreted to the extracellular environment participate in the pathogenesis and anthelmintic immune responses. The aim of this work was to identify and functionally annotate the excretory/secretory (ES) proteins of the monogenean ectoparasite *Rhabdosynochus viridisi* through bioinformatic approaches. A total of 1655 putative ES proteins were identified, 513 (31%) were annotated in the UniProtKB/Swiss-Prot database, and 269 (16%) were mapped to 212 known protein domains and 710 GO terms. We identified six putative multifunctional proteins. A total of 556 ES proteins were mapped to 179 KEGG pathways and 136 KO. ECPred predicted 223 enzymes (13.5%) and 1315 non-enzyme proteins (79.5%) from the secretome of *R. viridisi*. A total of 1045 (63%) proteins were predicted as antigen with a threshold 0.5. We also identified six venom allergen-like proteins. Our results suggest that ES proteins from *R. viridisi* are involved in immune evasion strategies and some may contribute to immunogenicity.

**Keywords:** Platyhelminthes; genomics; peptidase; bioinformatics; secretome; marine fish

## 1. Introduction

Monogeneans are ectoparasitic flatworms commonly found on the gills of marine and freshwater fish. Monogenean infections can cause excess mucosal secretion, epithelial damage, hemorrhage, osmotic problems, and gill atrophy leading to respiratory system failure [1]. In addition, lesions caused by monogenean parasites can facilitate secondary infections by bacteria, which can lead to the death of the fish host [2]. Some monogenean species belonging to the Diplectanidae family have been responsible for diseases and mortality in finfish aquaculture [3–6].

Several diplectanids of the genus *Rhabdosynochus* have been reported infecting the gill lamellae of wild and cultured snooks, *Centropomus* spp. (Perciformes: Centropomidae) [7,8]. Particularly, in northwestern Mexico, the diplectanid *Rhabdosynochus viridisi* has been reported causing mortality in a broodstock of Pacific white snook, *Centropomus viridis* [9]. This fish species is a well-suited candidate for marine cage aquaculture [10]. Additionally, snooks reared in laboratory and in a pilot commercial-scale farm have been affected by *R. viridisi* (Morales-Serna F.N. pers. obs.). Therefore, fish–parasite interactions should be better understood to support future research focused on the development of treatments to prevent or control monogenean infections in snooks.

Parasites secrete several biomolecules to ensure their entry and survival in the host. In particular, the proteins secreted to the extracellular environment, also known as secretome or excretory/secretory (ES) proteins, participate in the pathogenesis and immune

responses [11]. Although the secreted proteins could participate in the homeostasis of the parasite or its defense against other pathogens (e.g., part of the integument against bacteria), a critical role in the interaction with the host has been proposed. Some ES proteins induce apoptosis [12] or cause proteolysis in order to remodel tissues for invasion [13], and others act as immunomodulators to evade the host's immune system for parasite establishment and adaptation [14,15]. Immune evasion is a strategy used by pathogenic organisms to maximize their probability of transmission to a fresh host. It may involve (1) hiding from the immune system (e.g., antigen mimicry or masking, diversity/polymorphism, variation of parasite antigens); (2) interfering with the function of the immune system (e.g., blocking pattern recognition receptors (PRRs) and downstream signaling, inhibition of humoral factors); or (3) destroying elements of the immune system (e.g., induction of apoptosis of immune cells) [16–18].

The identification and analysis of ES proteins in parasitic flatworms have mainly been carried out in species of cestodes and trematodes that mostly infect mammals [19–22], whereas monogeneans have received less attention [23,24]. To further our understanding of monogeneans and their interaction with their fish hosts, we need better knowledge about their secreted proteins. Experimental studies focused on the extraction of these proteins in monogeneans may be challenging due to their small size (in the range of micrometers) and the difficulty of obtaining sufficient individuals to purify the required amounts of proteins. In this case, analysis in silico may be useful to predict ES proteins and to guide future experiments [25]. Thus, based on a recently released transcriptome of *R. viridisi* [26], the aim of the present work was to characterize in silico the ES proteins. We analyzed the possible roles of secretory antigens in immunity against this monogenean, with emphasis on recent developments regarding their immunomodulatory action and potential involvement in evasion of the host immune defense.

## 2. Results

### 2.1. In Silico Identification and Functional Annotation of ES Proteins

A total of 1655 non-redundant ES proteins from *R. viridisi* were predicted (Supplementary Materials). Only 513 proteins (31%) were annotated in the UniProtKB/Swiss-Prot database (Supplementary Materials) and 182 (11%) showed sequence similarity to proteins reported in WormBase Parasite (Supplementary Materials). Eighteen proteins (including three C-type lectins and five peptidases) showing high similarity to fish (Teleostei) and helminth (Lophotrochozoa) proteins were included in the secretome (Supplementary Table S1).

Of the 1655 predicted ES proteins, 269 (16%) were mapped to 212 known protein domains. The most represented protein domains were immunoglobulin, leucine-rich repeat domain (involved in protein–protein and protein–ligand interactions), and zinc finger (involved in protein–nucleic acid interactions), which are also present in non-ES proteins (Supplementary Materials). Lectin C-type domain (carbohydrate-binding), CAP domain, and serine and cysteine peptidase domains, related to immune-regulatory proteins, were also found. We identified six putative multifunctional proteins (Supplementary Table S2). Two of them, thioredoxin and thrombospondin 1, are included in MultitaskProtDB.

We mapped 269 (16%) ES proteins to 710 GO terms (229 Biological Process, 239 Cellular Component, and 242 Molecular Function categories). The most enriched GO terms (Figure 1) in *R. viridisi* were: cell (GO:0005623), cell part (GO:0044464), organelle (GO:0043226), membrane (GO:0016020), and membrane part (GO:0044425) at the Cellular Component category; anatomical structure development (GO:0048856) at the Molecular Function category, and cellular process (GO:0009987), cellular metabolic process (GO:0008152), and nitrogen compound metabolic process (GO:0006807) at Biological Process category. Similarly, we mapped 2419 (6.3%) non-ES proteins to 6324 GO terms (2057 Biological Process, 2133 Cellular Component, and 2134 Molecular Function categories). As shown in Figure 1, the GO terms with higher representation of ES proteins with respect to non-ES proteins were endomembrane system, cell periphery, plasma membrane, plasma

membrane part, membrane, membrane part, intrinsic component of membrane, extracellular region part, extracellular region, anatomical structure development, anatomical structure morphogenesis, and multicellular organism development (20% of proteins or more). More ES proteins than non-ES proteins were assigned to the GO terms cell surface, extracellular organelle, extracellular space, cell junction, regulation of developmental process, immune effector process, interspecies interaction between organisms, biological adhesion, and cell adhesion.

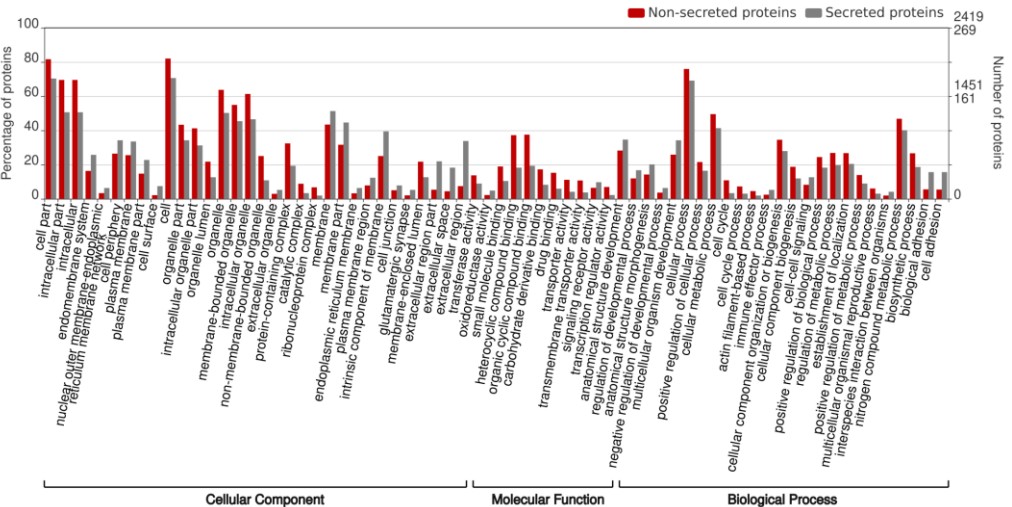

**Figure 1.** Gene ontology enrichment analysis of secreted and non-secreted proteins as compared to the transcriptome from *R. viridisi.* Only significantly (Pearson Chi-Square test *p*-value < 0.05) overrepresented GO terms are shown in this figure.

In total, 556 ES proteins were mapped to 179 KEGG pathways and 136 KO (Supplementary Materials). Several ES proteins were associated with human papillomavirus infection, proteoglycans and other pathways in cancer, and Alzheimer disease. These pathways were also present in non-ES proteins. Cathepsins B, C, and D were reported in the KEGG apoptosis pathway [PATH: ko04210]. KEGG BRITE objects related to membrane interactions (Membrane trafficking [BR:ko04131], Transporters [BR:ko02000]), were overrepresented in both ES and non-ES proteins. Thirty-four ES proteins were annotated as cell adhesion molecules (Supplementary Table S3).

ECPred predicted 223 enzymes (13.5%), 1315 non-enzyme proteins (79.5%), and 117 proteins without any prediction (7%) from the secretome of *R. viridisi* (Supplementary Materials). The putative enzymes were classified according to six classes of the Enzyme Commission (EC). Most of the predicted enzymes were represented by hydrolases (53%), followed by transferases (30%) and oxidoreductases (10%) (Figure 2A). Only 41% of the hydrolases were classified into EC subclasses (Figure 2B): 23% esterases (EC 3.1), 15% peptidases (EC 3.4), and 3% glycosylases (EC 3.2). Of the 1655 ES proteins, 12 were identified as Carbohydrate-active enzymes (CAZymes). They were included into nine different families (Supplementary Table S4) of the three major classes: five glycoside hydrolases, five glycosyltransferases, and two polysaccharide lyases. MEROPS analysis of the ES proteins resulted in the identification of 27 peptidases and 14 peptidase inhibitors (Supplementary Materials). Cysteine and serine peptidases were the most represented. The peptidase sub-families found in the secretome are shown in Figure 3. The I93 subfamily of peptidase inhibitors was the most abundant. ECPred predicted four glycosylases and 18 peptidases; however, dbCAN2 predicted five glycosylases and the alignment against the MEROPS database resulted in 27 peptidases in the secretome of *R. viridisi*.

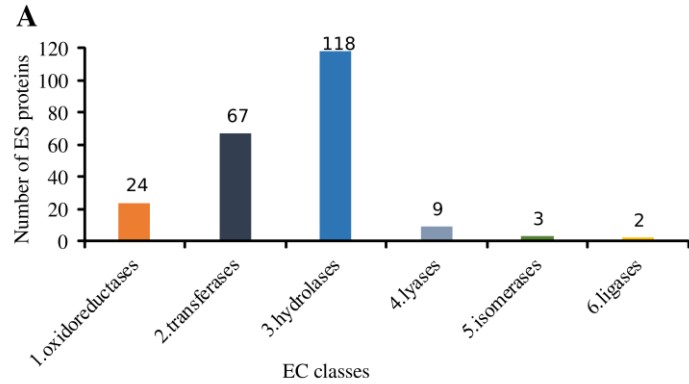

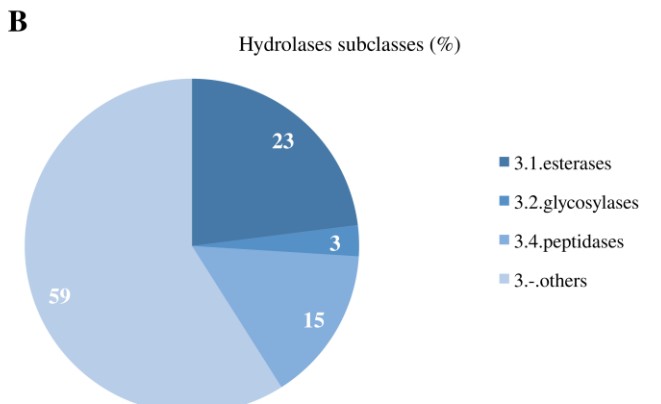

**Figure 2.** ES proteins of *R. viridisi* with enzymatic function predicted by ECPred: (**A**) Enzyme Commission (EC) classes, and (**B**) subclasses of hydrolases.

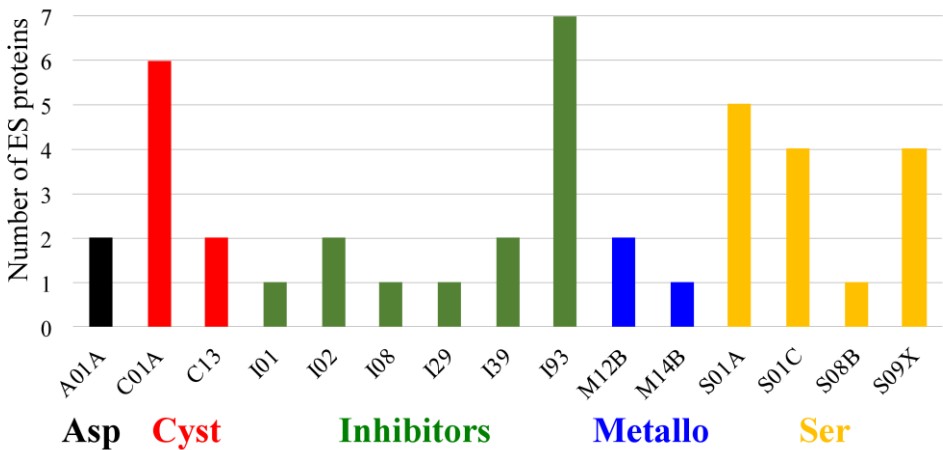

**Figure 3.** Peptidase families (Asp: aspartic peptidases, Cyst: cysteine peptidases, Inhibitors, Metallo: metallopeptidases, Ser: serine peptidases), and subfamily identifiers in predicted ES proteins.

## *2.2. Antigenicity Prediction of ES Proteins*

A total of 1045 (63%) ES proteins were predicted as antigens with a threshold of 0.5. Of these, 43 had a score higher than 0.9 (Supplementary Table S5). Of those, only six were functionally annotated (Collagen alpha-2(I) chain, GTPase NRas, Collagen alpha-2(IV) chain, Beta-1,4-galactosyltransferase 2, Inactive histone-lysine N-methyltransferase 2E, and Transcription factor 12). The proteins with unknown function were submitted to Phyre 2 and the Protein Data Bank (PDB) template name with higher confidence (>90%) and

percentage of identity (>25%) was selected. Table 1 shows the six most antigenic proteins and VAL proteins.

**Table 1.** The ES proteins of *R. viridisi* with potential immunomodulatory functions.

| Putative Immunomodulators | Protein ID | Annotation or PDB Template Name | Vaxijen Score |
|---|---|---|---|
| More antigenic proteins | DN725_c0_g1_i1.p1 | Collagen alpha-2(I) chain | 1.3983 |
| | DN18884_c0_g6_i2.p1 | Unknown: nigellin-1.1 | 1.3867 |
| | DN2184_c0_g1_i6.p6 | Unknown: fkbp-type peptidyl-prolyl cis-trans isomerase slyd | 1.3846 |
| | DN146924_c0_g3_i1.p1 | Unknown: SH3-like barrel | 1.2806 |
| | DN9007_c0_g1_i4.p2 | Unknown: protein phosphatase 1 regulatory subunit 3a | 1.2422 |
| | DN8473_c0_g5_i3.p1 | Unknown: ribosomal protein L14e | 1.2028 |
| Venom allergen-like (VAL) proteins | DN106_c0_g1_i4.p14 | GLIPR1-like protein 1 | 0.6280 |
| | DN1638_c0_g1_i10.p1 | Pathogenesis-related protein | 0.5865 |
| | DN1553_c0_g1_i1.p1 | Peptidase inhibitor 16 | 0.6481 |
| | DN2123_c1_g1_i5.p1 | Unknown: venom allergen-like protein 4 | 0.4954 |
| | DN2793_c0_g1_i1.p7 | Pre-mRNA-splicing factor ISY1 homolog | 0.4541 |
| | DN537_c0_g1_i4.p3 | Scoloptoxin SSD976 | 0.4269 |

## 3. Discussion

This is the first study predicting secreted proteins of a monogenean member of the Diplectanidae family. Previously, Caña-Bozada et al. [23] identified the ES proteins in four monogenean species, *Eudiplozoon nipponicum*, *Gyrodactylus salaris*, *Neobenedenia melleni*, and *Protopolystoma xenopodis*. Similarly, our present work suggests that various predicted ES proteins from *R. viridisi* are potentially involved in adhesion, penetration, incorporation of nutrients, and immunomodulation of host cells (Figure 4), which are functions with important roles in pathogenesis [14]. GO annotations of ES proteins from *R. viridisi* agree in most represented Molecular Functions and Biological Process categories with those previously reported for other helminths [25]. However, the annotation of membrane and membrane part in the Cellular Component category is only similar to other Platyhelminthes, including monogeneans [23]. For example, the secretion of cadherins, protocadherins, integrins, and collagen alpha chains (Supplementary Table S3) could be related to cell–cell adhesion after the physical insertion of haptors in the epithelial tissue, important to anchor and transient attachment for locomotion, achieved in monogeneans through cooperation between adhesive secretion [27] and the haptor [28].

The overrepresentation of hydrolases in the secretome of parasitic helminths has been reported elsewhere [25] and could be associated with proteolysis. Cytosolic cathepsins B and D could induce apoptosis in T cells [29]. These processes, proteolysis and apoptosis, contribute to the penetration of the parasite and destruction of host cells and immune receptors [17]. Putative cathepsins, essential for survival and presenting low homology to fish proteins, have been previously considered as potential drug targets due to their overrepresentation in ES proteins of other monogenean species [23]. We found the presence of cathepsins B, C, D, and L in both ES and non-ES proteins from *R. viridisi.* It has been reported that cathepsins L are important for nematodes and can degrade haemoglobin, serum albumin, immunoglobulins, fibronectin, collagen I, and laminin under acidic conditions, and its enzymatic activity is host-specific [30]. Also, cathepsins L have been characterized in the monogeneans *N. melleni* [31] and *E. nipponicum* [32]. *Eudiplozoon nipponicum* abundantly expresses cathepsins B, D, L1, and L3, which play a critical role in haemoglobin processing and immunomodulation [24]. *Rhabdosynochus viridisi*, like other mucus-feeding

monogeneans, could use elastase-like serine peptidases for extracellular digestion [32] instead of cathepsins, so it remains to verify the possible role of these peptidases in the secretome.

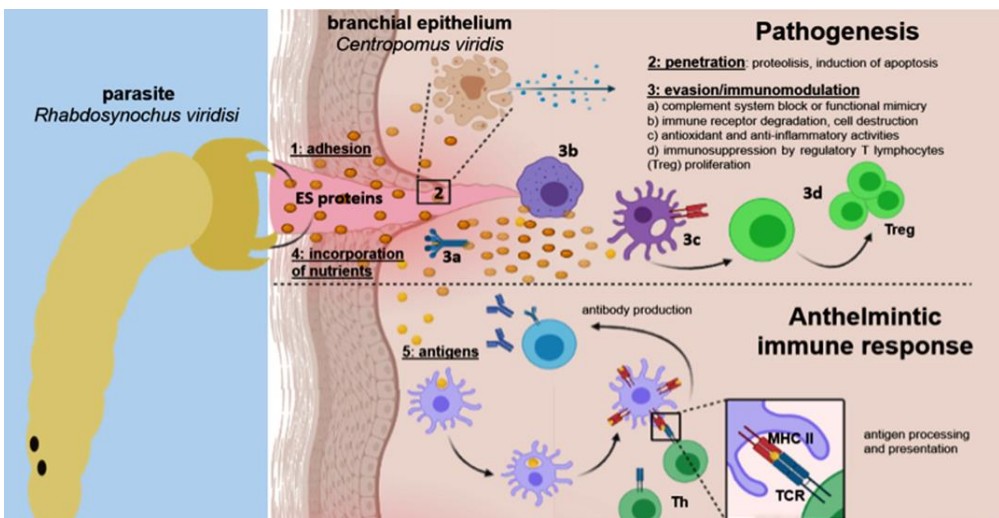

**Figure 4.** Excretory/secretory (ES) proteins from *Rhabdosynochus viridisi* could be involved in (1) adhesion, (2) penetration and destruction of branchial epithelium, (3) immune evasion strategies (in monocytes 3b, dendritic cells 3c, and T lymphocytes 3d), (4) nutrition, and (5) immunogenicity (the antigens are taken by Antigen Presenting Cells and then presented to T helper lymphocytes, which activate B cells for antibody production and secretion). Created in BioRender.com.

A predominance of inhibitors of the subfamily I93 was observed in the secretome of *R. viridisi*. This subfamily contains inhibitors of metallopeptidases that have been proposed to participate in extracellular matrix turnover, tissue remodeling, and other cellular processes in parasitic helminths [33]. In addition, the serpins (serine peptidase inhibitors I01, I02, I08) and cystatin (cysteine peptidase inhibitor I29) secreted by *R. viridisi*, could block complement activation, prevent inflammation, and induce immunosuppression by subverting Th1 mechanisms and drawing the immune system towards a Th2/Treg response, as has been reported for *E. nipponicum* [34,35]. Inhibitors of serine and cysteine peptidases, belonging to family I29, are highly transcribed and secreted by adult parasites of *E. nipponicum*, although they are not present in the secretome in large quantities [24].

The multifunctional proteins thioredoxin and thrombospondin 1 (TSP-1) found in the *R. viridisi* secretome could be related to pathogen virulence activity [36]. Thioredoxins of nematodes are ES proteins with antioxidant, anti-inflammatory, and anti-apoptotic activities. They also modify monocytes and epithelial cells, binding and inducing the time-dependent release of cytokines [37]. Thioredoxin has been also reported in the secretome of the monogenan *E. nipponicum* [24]. TSP-1 interacts with other extracellular matrix proteins to regulate cellular behavior. Moreover, it is secreted as a compensatory mechanism for controlling inflammation and protecting tissues from excessive damage [38]. TSP-1 has not been reported in monogeneans or other platyhelminths, but it is known that thrombospondin 2 (TSP-2) in the trematodes *Clonorchis sinensis* and *Fasciola hepatica* is a virulence and immunomodulation-related transcript that causes transforming growth factor beta stimulation [39]. In humans, TSP-1 and TSP-2 can interact with various ligands, such as structural components of the extracellular matrix, cytokines, cellular receptors, growth factors, proteases, and other stromal cell proteins [40]. This is the first report of thrombospondin 1 in a monogenean parasite.

The 18 predicted ES proteins from *R. viridisi* with high similarity to fish and helminths proteins possibly have mimicry function. For instance, Hebert et al. [41] observed that mimicry candidate proteins from the behavior-altering cestode *Schistocephalus solidus* had specific sequence similarity with proteins of the host (the threespine stickleback, *Gasteros-*

*teus aculeatus*). It has been suggested that a pathogen can mimic and substitute host proteins in order to hijack the host cellular processes [42]. Therefore, we would expect that potential mimicry proteins secreted by *R. viridisi* could be involved in immune-regulatory functions.

The most represented protein domains in the *R. viridisi* secretome were immunoglobulin (Ig), zinc finger, and leucine-rich repeats. Ig-like domain functions in *Caenorhabditis elegans* include cell–cell recognition, cell–surface receptors, muscle structure, and the immune system [43]. Zinc finger proteins participate in numerous physiological processes, such as cell proliferation, differentiation, and apoptosis, thereby maintaining tissue homeostasis. They are also implicated in transcriptional regulation, ubiquitin-mediated protein degradation, signal transduction, actin targeting, DNA repair, cell migration, and numerous other processes [44]. Particularly, zinc finger proteins are required for chromosome-specific pairing and synapsis during meiosis in *C. elegans* [45] and for survival of adult worms of *Schistosoma japonicum* [46]. The potential functions of zinc finger proteins in immune system regulation, both at the transcriptional and post-transcriptional levels, have been proposed recently [47]. Likewise, leucine-rich repeat domains are present in several immune receptors of animals and also participate in protein–protein interactions [48].

The most represented KEGG pathway, human papillomavirus (HPV) infection, could be associated with immune evasion. The HPV oncoproteins downregulate the expression of proinflammatory cytokines and chemokines, and upregulate the expression of immunosuppressive genes in host cells. They also degrade host proteins [49]; for example, we found the E3 ubiquitin-protein ligase UBR4 in the secretome, which seems to be involved in removing pro-apoptotic and pro-inflammatory molecules via the ubiquitin–proteasome system [50]. Several KEGG BRITE objects in ES proteins were related to membrane proteins suggesting a role in host–pathogen interactions.

Monogenean-specific antigens [51,52] and fish antibodies have been detected in several infections [53,54]. However, these studies are limited due to the difficulty of obtaining anti-Fc antibodies specific for different fish species, and high-quality protein extracts of small parasites, which are difficult to maintain in culture. The detection and identification of parasite antigens and humoral responses is important for vaccine development, and bioinformatic predictions have become essential. The present study suggests that most (63%) of the ES from *R. viridisi* are potential antigens and, therefore, could be processed and presented to T cell receptors, as well as recognized by host antibodies. According to the bioinformatic prediction of Gomez et al. [55], ES proteins of the platyhelminth *Taenia solium* are enriched in antigenic regions as compared to non-ES proteins. Antigens promoting antibody response could alleviate infection throughout antibody-dependent cellular cytotoxicity (ADCC) mechanisms [56]. For example, some structures of *Schistosoma* spp. and *Fasciola* spp., covered by antibodies, are destroyed by toxic proteins and reactive species released by cells with Fc receptors [57].

The most probable antigen predicted by Vaxijen was annotated as collagen alpha-2(I) chain. The collagen alpha-2(IV) chain also presented a score higher than 0.9. Recently, it was reported that human collagen alpha-2 type I stimulates collagen synthesis, wound healing, and elastin production in normal human dermal fibroblasts [58]. Type IV collagen, the most abundant constituent of the basement membrane, is highly conserved among vertebrates and invertebrates, regulating cell adhesion and migration [59]. Other most probable antigens are apparently unique to the species *R. viridisi*, since they did not have functional annotation by sequence homology. Species-specific antigens are common to members of a single species that improve the efficiency of immunological diagnosis and immunoprophylaxis of helminthic diseases [60]. Further research to identify antigens from the ES proteins in the tegument of monogeneans could shed light on the development of better strategies to prevent or control fish diseases. These antigens, such as membrane or tegumental proteins, have been proposed as vaccine candidates due to the probability of interaction with the host's immune system [61–64].

VAL proteins were also identified. These proteins are generally secreted in parasitic stages [65] and there is an increasing interest in characterizing their immunological

properties to design anthelmintic vaccines [66–68]. They are ubiquitous ES products that paralyze plants and animals, and have been previously named as sperm-coating protein/Tpx/antigen 5/pathogenesis-related-1/Sc (SCP/TAPS), cysteine-rich secretory proteins/antigen 5/pathogenesis-related 1 (CAP), or activation-associated secreted proteins (ASPs); the CAP domain seems to be conserved in evolution, allowing the binding of small hydrophobic ligands, but little is known regarding endogenous ligands [65]. There are no experimental studies of VAL proteins from monogenean species, and only 41 VAL transcripts from *N. melleni* have been reported [69]. Of the six VAL proteins of *R. viridisi*, only the peptidase inhibitor 16 (PI16) showed similarity to unnamed protein products from *P. xenopodis*, which is another monogenean species. PI16 suppresses the chemokine chemerin activation [70], which impairs endothelial cell inflammation [71]. GLIPR1-like protein 1 has been reported in ES from other parasites, suggesting a role in cell adhesion [72,73]. Pathogenesis-related proteins form a protective barrier against invasive pathogens, at least in plants [74]. Pre-mRNA-splicing factor ISY1 homolog is a component of the spliceosome C complex required for the selective processing of microRNAs (miRNAs) during embryonic stem cell differentiation. It is also involved in pre-mRNA splicing in the nucleus [75]. Scoloptoxin SSD976 is a voltage-gated calcium channel inhibitor [76] that could impair $Ca^{2+}$ signaling, T cell activation, and proliferation in fish [77]. Venom allergen-like protein 4 (VAL-4) has been reported as secreted by other helminths. VAL-4 presents lipid-binding properties and can sequester small hydrophobic ligands; this could be a mechanism to modulate immune responses in the host [65]. It remains to be determined if there is functional homology of these VALs in the context of monogenean–fish interaction, at least to verify their role in the parasite life cycle and modulation of the host.

The present study provides an in silico characterization of the secretome of *R. viridisi*; nevertheless, it has some limitations. The identified ES proteins of *R. viridisi* depend on the quality of the annotated transcriptome, and it is possible that proteins presenting high homology to fish belong to the host. Moreover, there is a lack of data from monogenean proteins, and annotation based only on sequence homology is not totally reliable. It is necessary to verify the presence of the secreted proteins through a proteomic approach, and to identify key virulence factors by functional studies, which are limited due to the size and difficulty of handling *R. viridisi* specimens. Antigenicity prediction is based on a human host, whose immune system is different from fish; the latter is composed of low-affinity antibodies, while antigen presentation and ADCC remain unverified.

## 4. Materials and Methods

Protein sequences were retrieved from an available *R. viridisi* transcriptome [26]. The specimens used for that transcriptome were all adults isolated from the gills of their fish host *C. viridis*, reared in laboratory conditions [26]. The predicted ES proteins were identified following the bioinformatic workflow of Gahoi et al. [25], which filters sequences based on signal peptide, discarding sequences with transmembrane regions, subcellular localization of mitochondrial proteins, endoplasmic reticulum retention signal, and GPI-anchor proteins.

To rule out contaminant sequences, the ES proteins were aligned using BLASTP against the NCBI non-redundant protein database (E-value $< 1 \times 10^{-5}$). The sequences that were best hits with Protostomia (taxid: 33,317) were retrieved, and the remaining sequences were considered contaminants. Proteins with similar E-value among Teleostei, Lophotrochozoa, and Platyhelminthes bases were included. ES proteins were annotated based on Uniprot-SwissProt and WormBase Parasite [78] databases, using BLASTP (E-value $< 1 \times 10^{-4}$). ES proteins without any homologue in the NCBI non-redundant protein database were considered as the specific secretome of *R. viridisi*.

The domains were identified with HMMSCAN v3.1.b2 [79] using the Pfam database as reference [80]. Possible multifunctional proteins (with two or more domains associated with different biochemical functions) were identified using the domain annotation and the Multitask ProtDB-II protein database [81]. The Gene Ontology (GO) terms were retrieved

with the PANNZER2 server [82] and were subsequently plotted and analyzed with the WEGO 2.0 server [83] by Pearson Chi-Square test, using the entire proteome as the reference group. A significant enrichment was considered when the *p*-value < 0.05. The GO terms were updated with QuickGO [84]. KEGG Orthology (KO) IDs and KEGG pathways were retrieved from the Trinotate annotation [26].

Enzymes were classified with the ECPred v1.1 software [85], using the option "weighted", according to the Enzyme Commission (EC) nomenclature based on the type of catalyzed reaction: oxidoreductases (EC 1), transferases (EC 2), hydrolases (EC 3), lyases (EC 4), isomerases (EC 5), and ligases (EC 6). CAZymes were identified with the dbCAN2 server [86]. Peptidases and peptidase inhibitors were identified by aligning ES proteins against the MEROPS database (merops_scan.lib) [87], using BLASTP (E-value $< 1 \times 10^{-4}$). Antigenic proteins were predicted with VaxiJen 2.0 server [88]. Those proteins with score higher than 0.9 were subjected to a fold recognition analysis with Phyre2 [89] for functional information. Venom allergen-like (VAL) proteins, which have cysteine-rich secretory proteins, antigen 5, and pathogenesis-related 1 (CAP) domain, were also analyzed with VaxiJen.

## 5. Conclusions

The present work represents hypothetical molecular mechanisms of host–parasite interactions of the monogenean *R. viridisi* with its host. We identified potential proteins secreted by this parasite and discussed their possible contribution to the pathogenesis and anthelmintic responses. Using bioinformatic analysis, we found that most of the ES from *R. viridisi* are potential antigens. This is the second study reporting the presence of VAL transcripts in a monogenean parasite. Further research is necessary to confirm the presence and function of these ES proteins in vivo.

**Supplementary Materials:** The following supporting information can be downloaded at: https://www.mdpi.com/article/10.3390/parasitologia3010004/s1; Supplementary File S1: Predicted ES proteins of *Rhabdosynochus viridisi*; Supplementary File S2: Annotation of ES proteins of *R. viridisi* based on Uniprot-SwissProt database; Supplementary File S3: Annotation of ES proteins of *R. viridisi* based on WormBase Parasite database; Supplementary File S4: Annotation of ES and non-ES proteins of *R. viridisi* based on Pfam domains; Supplementary File S5: Annotation of ES and non-ES proteins of *R. viridisi* based on KEGG pathways; Supplementary File S6: Annotation of ES proteins of *R. viridisi* based on ECPred; Supplementary File S7: Annotation of ES proteins of *R. viridisi* based on MEROPS database. Supplementary Tables file, containing Table S1: Proteins with similarity between fish and parasites; Table S2: Possible multifunctional proteins found in predicted ES proteins of *R. viridisi*; Table S3: Adhesion proteins identified in *R. viridisi*; Table S4: Families of carbohydrate-active enzymes (CAZymes) in predicted ES proteins of *R. viridisi*; Table S5: Most antigenic ES proteins from *R. viridisi*.

**Author Contributions:** Conceptualization, F.N.M.-S. and M.M.-C.; methodology, M.M.-C. and V.H.C.-B.; data curation, M.M.-C. and V.H.C.-B.; writing—original draft preparation, M.M.-C. and A.G.-G.; supervision, A.G.-G.; writing—review and editing, M.M.-C., A.G.-G., V.H.C.-B. and F.N.M.-S.; funding acquisition, F.N.M.-S. All authors have read and agreed to the published version of the manuscript.

**Funding:** This research was funded by the National Council of Science and Technology of Mexico (CONACYT) through the grant FORDECYT-PRONACES CF/1715616 awarded to FNMS. The APC was funded by AGG.

**Institutional Review Board Statement:** Not applicable. No animals were used in this study.

**Informed Consent Statement:** Not applicable.

**Data Availability Statement:** The data presented in this study are available inSupplementary Materials.

**Acknowledgments:** The authors would like to thank the National Council of Science and Technology (Mexico) for doctoral scholarships granted to M.M.-C. and V.H.C.-B.

**Conflicts of Interest:** The authors declare no conflict of interest.

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
