# Peer review of "Predicted Secretome of the Monogenean Parasite Rhabdosynochus viridisi: Hypothetical Molecular Mechanisms for Host-Parasite Interactions"

_parasitologia, doi:10.3390/parasitologia3010004_

Round 1

Reviewer 1 Report

This was a well-conducted and well-presented piece of work.

One suggestion I would like to see addressed is more detail in the Discussion regarding the potential uses of the information e.g. potential control options.  Obviously this would be very difficult for fish parasites, but should be discussed.

Author Response

We want to thank all reviewers for their constructive comments and suggestions. We have provided point-by-point answers to all the comments and hopefully we answered the questions. All modifications are highlighted in yellow in the main manuscript; all supplementary files were corrected; all Tables (except Table 4, now Table 1) became supplementary Tables; Figure 1 was corrected, and a graphical abstract was included.

Reviewer 1

This was a well-conducted and well-presented piece of work.

One suggestion I would like to see addressed is more detail in the Discussion regarding the potential uses of the information e.g. potential control options.  Obviously this would be very difficult for fish parasites, but should be discussed.

Thank you for your suggestion. We added these lines to the Discussion:

 “Putative cathepsins, essential for monogenean survival and presenting low homology to fish proteins, have been previously considered as potential drug targets [20]…”

“…Further research to identify antigens from the ES proteins in the tegument of monogeneans could shed light on the development of better strategies to prevent or control fish diseases. These antigens, such as membrane or tegumental proteins, have been proposed as vaccine candidates due to the probability of interaction with the host's immune system [45-48].”

Reviewer 2 Report

A major flaw in this manuscript is that there does not appear to be any new information and the study is rather incomplete. 

It appears the authors duplicated an approach from reference 19. The current paper does not appear to add any new insight from the previous study [19]. Perhaps the use of a monogenean parasite is the new information. But that was never made clear. If so, the authors should discuss in detail how their data is similar or different than the previous study [19]. 

The authors also conclude they have identified potential immunomodulatory proteins. Yes, E/S proteins of helminthes modulate the immune system. But that is already known and this study does not contribute to our further understanding of this phenomenon. 

One thing that is clearly lacking in this manuscript, as acknowledged by the authors, is experimental evidence about the 'hypothetical' E/S proteins. Some proteomic data would add value to the paper. 

Even without proteomic data there are many well characterized E/S proteins. Are any of the hypothetical E/S proteins of Rhabdosynochus homologes of well known E/S proteins? That would certainly be useful information to include in the Results and Discussion, and could strengthen claims about the functions of E/S proteins. 

The data is primarily an analysis of the hypothetical E/S proteins in regards to KO, KEGG, ECPred, etc. However, there is no information provided about the non-E/S proteins in regards to KO, KEGG, ECPred, etc. Knowing the differences between E/S proteins and non-E/S proteins may add some value to the data. 

It appears that the authors do not know what antigenicity means. Or at least its use in line 27 and line 186 appears inappropriate. What do the authors mean by antigenicity in the context of the E/S proteins and why is this significant. The conventional use of antigenicity refers to the interactions between antigens and B- and T-cell receptors. Do the authors mean immunogenicity? 

Likewise, section 2.2 and Table 4 are not clear at all. Why is it important that a computer program predicted epitopes (based on human immune system) in many of the E/S proteins? What about epitope prediction in non-E/S proteins? And VAL needs a lot more description and discussion. 

Table 1 is also not clear in regards to the Pfam and Description columns. There needs to be some alignment between the various Pfams and their descriptions. In addition, what is significant about the 10 most represented? Could it be the 5 most represented or the 15 most represented? Same comment for Tables 2 and 3. 

Some description about the contents of the supplementary files would be nice. File 1 is a FASTA file and will not be widely assessible. The spreadsheets are assessible, but there is no description of the columns and what the data represents. Descriptions would certianly make the supplementary material more useful. 

Line 195. What is the relationship between T. cruzi and R viridisi? Seems like meaningless trivia. 

Line 163. What does 'were annotated' mean?

Figure 4 is probably not necessary. 

The paragraph at lines 206-225 seems to be a lot of unfounded speculation. 

Line 243. The Phrye2 data was never shown. 

Line 280. Reference 62. The link on line 499 is wrong. Could not find this paper. 

Author Response

We want to thank all reviewers for their constructive comments and suggestions. We have provided point-by-point answers to all the comments and hopefully we answered the questions. All modifications are highlighted in yellow in the main manuscript; all supplementary files were corrected; all Tables (except Table 4, now Table 1) became supplementary Tables; Figure 1 was corrected, and a graphical abstract was included.

Reviewer 2

1) A major flaw in this manuscript is that there does not appear to be any new information and the study is rather incomplete. 

It appears the authors duplicated an approach from reference 19. The current paper does not appear to add any new insight from the previous study [19]. Perhaps the use of a monogenean parasite is the new information. But that was never made clear. If so, the authors should discuss in detail how their data is similar or different than the previous study [19]

Thank you for the observation. Regarding reference 19, we certainly used a similar bioinformatic approach for obtaining ES proteins. The authors used protein sequences of 73 nematode genomes, downloaded from the wormbase database as of June 30, 2016. They also retrieved EST sequences from NCBI dbEST database as of August 11, 2016. By then, only the monogenean Gyrodactylus salaris genome was available. To date, four monogenean genomes are available in NCBI (G. salaris, G. bullatarudis, Protopolystoma xenopodis, and Benedenia humboldti) and transcriptomic data of Eudiplozoon nipponicum. De novo-assembled transcriptomes of Scutogyrus longicornis and Rhabdosynochus viridisi are a recent contribution to the monogenean data. Thus, even if it is a similar approach, genomic information regarding monogenean species changed from 2016 to date. Moreover, a novel contribution respect reference 19 is the Antigenicity prediction of ES proteins.

It is important to mention that this is the first study focused on the ES proteins of a member of the Diplectanidae family. To make this clearer, we added this sentence to the Introduction:

“…This is the first study focused on the ES proteins of a member of the Diplectanidae family”.

In addition, in the Discussion section we added the following phrase in the manuscript:

“Our study adds new insights regarding ES proteins and predicted functions in a fish monogenean parasite.”

2) The authors also conclude they have identified potential immunomodulatory proteins. Yes, E/S proteins of helminthes modulate the immune system. But that is already known and this study does not contribute to our further understanding of this phenomenon.

The reviewer is right; immunomodulatory functions of ES proteins have been previously reported. We searched the literature for hypothetical molecular mechanisms for host-parasite interactions in the context of a monogenean parasite and a fish. Most of the studies focus on mammals, particularly humans, and parasite species like Trematoda and Cestoda. These predictions can lead researchers for future experiments with fish and monogenean parasites. There is a lack of knowledge in monogenean ES proteins (in our study just 31 % were annotated) and also the immune system of fish. For example, we found that R. viridisi may secrete scoloptoxin, and if this is verified by proteomic approaches in future studies, it will be interesting to know if the use of scoloptoxin inhibitors results in better fish defenses.  Also, the lack of fish vaccines against parasites could be due (at least in part) to the lack of knowledge regarding “virulence” proteins in parasites and the production of “resistance” proteins in the host.

We agree that our work may not provide a novel contribution for the helminthology in general, but it provides novel information for a particular group of parasitic platyhelminths (monogeneans), typically underrepresented in molecular studies. 

3) One thing that is clearly lacking in this manuscript, as acknowledged by the authors, is experimental evidence about the 'hypothetical' E/S proteins. Some proteomic data would add value to the paper. 

The reviewer is right. We acknowledge this limitation in the manuscript:

“It is necessary to verify the presence of the secreted proteins through a proteomic approach, and to identify key virulence factors by functional studies, which are limited due to the size and difficulty to handle R. viridisi specimens.”

Unfortunately, the generation of proteomic data is currently out of our possibilities. It is challenging (although not impossible) to collect good quality parasite samples in sufficient quantities for proteomic analysis.

4) Even without proteomic data there are many well characterized E/S proteins. Are any of the hypothetical E/S proteins of Rhabdosynochus homologes of well known E/S proteins? That would certainly be useful information to include in the Results and Discussion, and could strengthen claims about the functions of E/S proteins. 

Thank you for the observation. There are indeed hypothetical ES proteins found in R. viridisi with well-known functions. In the discussion section we now emphasized that R. viridisi has several well-known homologs such as cathepsins B and D, found in other monogeneans like Eudiplozoon nipponicum. Other examples mentioned in this work are serpins (serine peptidase inhibitors), cystatin (cysteine peptidase inhibitor), and thioredoxin.

5) The data is primarily an analysis of the hypothetical E/S proteins in regards to KO, KEGG, ECPred, etc. However, there is no information provided about the non-E/S proteins in regards to KO, KEGG, ECPred, etc. Knowing the differences between E/S proteins and non-E/S proteins may add some value to the data.

Thank you for your comment. Our work focuses on the predicted secretome as an important component to predict host-parasite interactions between fish and monogenean ectoparasites. We agree that non-ES proteins are important, however the identification and analysis of these non-ES proteins is outside of the aim of the present study.

6) It appears that the authors do not know what antigenicity means. Or at least its use in line 27 and line 186 appears inappropriate. What do the authors mean by antigenicity in the context of the E/S proteins and why is this significant. The conventional use of antigenicity refers to the interactions between antigens and B- and T-cell receptors. Do the authors mean immunogenicity? 

The reviewer is right. We changed the text as follows:

“…some may contribute to immunogenicity”, and line 186 (Figure 4 legend) to “immunogenicity”

We believe that the “antigenic proteins” found in the secretome may contribute to the “immunogenicity” (anthelmintic response). 

7) Likewise, section 2.2 and Table 4 are not clear at all. Why is it important that a computer program predicted epitopes (based on human immune system) in many of the E/S proteins? What about epitope prediction in non-E/S proteins? And VAL needs a lot more description and discussion. 

Thank you for the observation. It is true that Vaxijen is based on the human immune response. However, this kind of tool is not available for fish. Currently, several studies dealing with immunoinformatics in fish use Vaxijen. We used Vaxijen to get an idea of the probability of ES proteins to be recognized by the fish immune system.

VAL proteins have been well described in helminthes, and we have added more information to the manuscript: “…They are ubiquitous ES products that paralyze plants and animals, and have been previously named as sperm-coating protein/Tpx/antigen 5/pathogenesis-related-1/Sc (SCP/TAPS), cysteine-rich secretory proteins/antigen 5/pathogenesis-related 1 (CAP), or activation-associated secreted proteins (ASPs); the CAP domain seems to be conserved in evolution, allowing the binding of small hydrophobic ligands, but little is known regarding endogenous ligands [49]…” “…Venom allergen like protein 4 (VAL-4) has been reported as secreted by other helminthes; VAL-4 presents lipid-binding properties and can sequester small hydrophobic ligands; this could be a mechanism to modulate immune responses in the host [49].” 

8) Table 1 is also not clear in regards to the Pfam and Description columns. There needs to be some alignment between the various Pfams and their descriptions. In addition, what is significant about the 10 most represented? Could it be the 5 most represented or the 15 most represented? Same comment for Tables 2 and 3. 

Thank you for the observations. Basically, top 10 was a matter of space. In the revised manuscript, we decided to move the Tables to the Supplementary material. All identified domains, KEGG pathways and KEGG BRITE were now included.

9) Some description about the contents of the supplementary files would be nice. File 1 is a FASTA file and will not be widely assessible. The spreadsheets are assessible, but there is no description of the columns and what the data represents. Descriptions would certianly make the supplementary material more useful. 

Thank you for the suggestion. We added some descriptions in the supplementary file section in order to improve the understanding and use of supplementary files.

  • Supplementary File 1: Predicted ES proteins of Rhabdosynochus viridisi
  • Supplementary File 2: Annotation of ES proteins of viridisi based on Uniprot-SwissProt database
  • Supplementary File 3: Annotation of ES proteins of viridisi based on WormBase Parasite database
  • Supplementary File 4: Annotation of ES proteins of viridisi based on KEGG pathways
  • Supplementary File 5: Annotation of ES proteins of viridisi based on ECPred
  • Supplementary File 6: Annotation of ES proteins of viridisi based on MEROPS database
  • Supplementary Table S1. Proteins with similarity between fish and parasites. E-values were obtained from the alignment of the sequences of viridisi against sequences from different taxa using the NCBI database.
  • Supplementary Table S2. Pfam domains in ES proteins from viridisi.
  • Supplementary Table S3. Possible multifunctional proteins found in predicted ES proteins of viridisi.
  • Supplementary Table S4. KEGG pathways in ES proteins from viridisi.
  • Supplementary Table S5. KEGG BRITE objects in ES proteins from viridisi.
  • Supplementary Table S6. Adhesion proteins identified in viridisi.
  • Supplementary Table S7. Families of carbohydrate-active enzymes (CAZymes) in predicted ES proteins of viridisi.
  • Supplementary Table S8. Most antigenic ES proteins from viridisi (VaxiJen score > 0.9)

10) Line 195. What is the relationship between T. cruzi and R viridisi? Seems like meaningless trivia

Both T. cruzi and R. viridisi are unrelated (phylogenetically distant parasites). However, as information about ES proteins is scarce for platyhelminths, we used available information for other parasite groups in order to make some comparisons. The reason was to confirm that our approach allows identifying ES proteins already known for other parasites. In this case, we refer to T. cruzi because is a parasite in which thrombospondin has been also found.

11) Line 163. What does 'were annotated' mean?

We changed “were annotated” for: “…were potentially involved in…”

12) Figure 4 is probably not necessary. 

The reviewer is right, however the figure summarizes our findings and (if the reviewer agrees) we would like to leave it.

13) The paragraph at lines 206-225 seems to be a lot of unfounded speculation

Thank you for the comment. In this paragraph we are trying to explain part of the results of functional annotation based on what is known for other organisms, but in the context of parasitism.

14) Line 243. The Phrye2 data was never shown. 

All proteins with unknown annotation were submitted to Phyre 2, and we selected the most probable outcome. For example, unknown protein ID DN18884_c0_g6_i2.p1 was reported as nigellin-1.1 by Phyre 2.

We added the following paragraph in the Results section:

“The proteins with unknown function were submitted to Phyre 2 and the Protein Data Bank (PDB) template name with higher confidence (> 90 %) and percentage of identity (> 25 %) was selected.”

15) Line 280. Reference 62. The link on line 499 is wrong. Could not find this paper. 

The reviewer is correct. The link is now available.

Reviewer 3 Report

Short Summary
The work here present of Mirabent-Casals et al. is an in silico descriptional analysis of the excretory/secretory proteins of the monogenean Rhabdosynochus viridisi based on previous available transcriptomic data. The work is clear in its objectives and show interesting information about these kind of proteins present in the parasite. There are no experimental data in addition to the bioinformatics analysis that is shown here. I have some suggestions and comments for the authors.

Language
The language is correct.

Major points
- I understand that it is an in silico work. Still, it is worth it that you show here what are the most relevant points of your work, which are the things that are more important to corroborate in future experimental approaches, if it is difficult in this parasite, could some experiments be done in another model species? This will help to improve the manuscript in general, in particular in the discussion.
- I suggest comparing your results with more available information on other monogenean species, there is a bit discussed here, but I considered that this should be extended a bit more.

Minor points
-Lines 103 to 105: " Several ES proteins were associated with human papillomavirus infection, proteoglycans and other pathways in cancer, and Alzheimer disease, which involve inhibition of apoptosis and immune evasion." Did you check which proteins are here associated with these pathways? Are these proteins involved in apoptosis or immune evasion?

-Lines 115-116: "ECPred predicted 223 enzymes (13.5 %) and 1315 non-enzyme proteins (79.5 %) from the secretome of R. viridisi", and the other 7%?

- Figure 3: I do not see in the legend what is in the figure; correlate better the names with the description here.

-Table 4. This table should be explained better in the text. Some of the Proteins have annotation, and others said Unknown, I imagine that the ones without the "Unknown" are clearly that protein by your analysis? Others do not say anything (DN706_c0_g1_i1.p1 and DN137_c0_g1_i5.p1), why are they just with a blank space?

- Table 4. "Table 4 shows the six most antigenic proteins and VALs proteins." But there are shown 7 Protein IDs and after that the VALs. Is this correct?

- Figure 4. In the legend, there are not present all the references for the different names of the cells and proteins present in the figure.

- Lines 220-2021: "The most represented KEGG pathway, human papillomavirus (HPV) infection, 220 could be associated with immune evasion." Are the proteins here associated here involved in immune evasion?

- Lines 244-248: "Further research to identify tegumental antigens in monogeneans could shed light on the development of better strategies to prevent or control fish disease. It is because antigens found on the surface of pathogens, such as membrane or tegumental proteins, have been analyzed and proposed as vaccine candidates due to their high probability of interaction with the host's immune system [45-48]."  Agree, but, it is not clear the relation of the ES proteins with this part of the text here.

- Line 280: "Protein sequences were retrieved from an available R. viridisi transcriptome [62]." Mention a bit more information about these data, in particular the parasites, and early or late infection. That is important to understand the data here presented.

Author Response

We want to thank all reviewers for their constructive comments and suggestions. We have provided point-by-point answers to all the comments and hopefully we answered the questions. All modifications are highlighted in yellow in the main manuscript; all supplementary files were corrected; all Tables (except Table 4, now Table 1) became supplementary Tables; Figure 1 was corrected, and a graphical abstract was included.

Reviewer 3

1) I understand that it is an in silico work. Still, it is worth it that you show here what are the most relevant points of your work, which are the things that are more important to corroborate in future experimental approaches, if it is difficult in this parasite, could some experiments be done in another model species? This will help to improve the manuscript in general, in particular in the discussion.

Thank you for your suggestion. We added the following information in the manuscript:

“…This is the first study focused on the ES proteins of a member of the Diplectanidae family...”

“…Our study adds new insights regarding ES proteins and predicted functions in a fish monogenean parasite...”

“…It is necessary to verify the presence of the secreted proteins through a proteomic approach, and to identify key virulence factors by functional studies, which are limited due to the size and difficulty to handle R. viridisi specimens...”

  We think that new techniques for obtaining several specimens in culture will allow future proteomic analysis in order to corroborate the presence of secreted proteins. Also, recombinant protein expression and/or protein purification techniques will improve the functional characterization of these ES proteins in vitro and in vivo.

2) I suggest comparing your results with more available information on other monogenean species, there is a bit discussed here, but I considered that this should be extended a bit more.   Thank you very much for the suggestion. The information regarding monogean species is scarce. To date, only four monogenean genomes are available in NCBI (Gyrodactylus salaris, G. bullatarudis, Protopolystoma xenopodis, and Benedenia humboldti) and transcriptomic data of Eudiplozoon nipponicum. De novo-assembled transcriptomes of Scutogyrus longicornis and Rhabdosynochus viridisi are a recent contribution to the monogenean data. Our results are similar to those reported in reference [20], but we are adding new information regarding to the VAL proteins. There are no experimental studies of VAL proteins from monogenean species, and only 41 VAL transcripts from Neobenedenia melleni have been reported [53].   Nevertheless, we were not able to access to the transcriptomic data of Neobenedenia mellenito compare our results.    3) Lines 103 to 105: " Several ES proteins were associated with human papillomavirus infection, proteoglycans and other pathways in cancer, and Alzheimer disease, which involve inhibition of apoptosis and immune evasion." Did you check which proteins are here associated with these pathways? Are these proteins involved in apoptosis or immune evasion?   Thank you for the observation. We removed the following phrase: “which involve inhibition of apoptosis and immune evasion.”

To try to answer the question, we only found one ES protein in the human papillomavirus infection pathway, the E3 ubiquitin-protein ligase UBR4, which has been reported to be involved in apoptosis or immune response:

“Indeed, when the UBR1, UBR2, UBR4, and UBR5 were knocked down using RNAi, the secretion of IL-1β was significantly increased [134], suggesting that the degradation of these proinflammatory fragments via the Arg/N-degron pathway plays a vital regulatory role in inflammatory responses…”

“…pro-apoptotic fragments have acquired N-degrons directly from proteolytic cleavage or through the actions of ATE 1, which are recognized by E3 N-recognins and removed via the ubiquitin-proteasome system.”

Kim, J.G.; Shin, H.-C.; Seo, T.; Nawale, L.; Han, G.; Kim, B.Y.; Kim, S.J.; Cha-Molstad, H. Signaling Pathways Regulated by UBR Box-Containing E3 Ligases. Int. J. Mol. Sci. 2021, 22, 8323. https://doi.org/10.3390/ijms22158323

Also, ADAM10, an ES protein from the Alzheimer disease pathway, is related with apoptosis:

“ADAM10 as an important regulator of CD46 expression during apoptosis. The ADAM10-mediated release of CD46 from apoptotic vesicles may represent a form of strategy to allow restricted complement activation to deal with modified self.”

Hakulinen J, Keski-Oja J. ADAM10-mediated release of complement membrane cofactor protein during apoptosis of epithelial cells. J Biol Chem. 2006 Jul 28;281(30):21369-21376. doi: 10.1074/jbc.M602053200.

“ADAM10 represents an important molecular modulator of FasL-mediated cell death.”

Schulte, M., Reiss, K., Lettau, M. et al. ADAM10 regulates FasL cell surface expression and modulates FasL-induced cytotoxicity and activation-induced cell death. Cell Death Differ 14, 1040–1049 (2007). https://doi.org/10.1038/sj.cdd.4402101   4) Lines 115-116: "ECPred predicted 223 enzymes (13.5 %) and 1315 non-enzyme proteins (79.5 %) from the secretome of R. viridisi ", and the other 7%?   Thank you for the observation. The other 7% was included in “no prediction” by the software ECPred. This was indicated in the document.   5) Figure 3: I do not see in the legend what is in the figure; correlate better the names with the description here.   We changed the legends for better understanding. Thank you.   6) Table 4. This table should be explained better in the text. Some of the Proteins have annotation, and others said Unknown, I imagine that the ones without the "Unknown" are clearly that protein by your analysis? Others do not say anything (DN706_c0_g1_i1.p1 and DN137_c0_g1_i5.p1), why are they just with a blank space?   The reviewer is right. We removed the codes DN706_c0_g1_i1.p1 and DN137_c0_g1_i5.p1 from the Table (now Table 1) because both had the same annotation: GLIPR1-like protein 1. Idem with DN725_c0_g1_i2.p1 that is the Collagen alpha-2(I) chain.

In Table 1 (previous Table 4), all proteins with unknown annotation were submitted to Phyre 2, and we selected the most probable outcome. For example, unknown protein ID DN18884_c0_g6_i2.p1 was reported as nigellin-1.1 by Phyre 2. We added the following lines:

“The proteins with unknown function were submitted to Phyre 2 and the Protein Data Bank (PDB) template name with higher confidence (> 90 %) and percentage of identity (> 25 %) was selected.”   7) Table 4. "Table 4 shows the six most antigenic proteins and VALs proteins." But there are shown 7 Protein IDs and after that the VALs. Is this correct?   Thank you for the observation. We corrected Table 4, now Table 1.   8) Figure 4. In the legend, there are not present all the references for the different names of the cells and proteins present in the figure.   Thank you for the observation. The following information was included in the legend: “Figure 4. Excretory/secretory (ES) proteins from Rhabdosynochus viridisi could be involved in 1) adhesion, 2) penetration and destruction of branchial epithelium, 3) immune evasion strategies (in monocytes 3b, dendritic cells 3c, and T lymphocytes 3d), 4) nutrition, and 5) immunogenicity (the antigens are taken by Antigen Presenting Cells and then presented to T helper lymphocytes, which activate B cells for antibody production and secretion). Created in BioRen-der.com”   9) Lines 220-221: "The most represented KEGG pathway, human papillomavirus (HPV) infection, 220 could be associated with immune evasion." Are the proteins here associated here involved in immune evasion?   We only found one ES protein in the human papillomavirus infection pathway, the E3 ubiquitin-protein ligase UBR4, which has been reported as involved in the degradation of pro-inflammatory fragments such as interleukin 1 beta and could also be associated with the immune receptors degradation. Many pathogens, mostly viruses, utilize mechanisms to target and destroy immune receptors in an attempt to evade immune responses.    10) Lines 244-248: "Further research to identify tegumental antigens in monogeneans could shed light on the development of better strategies to prevent or control fish disease. It is because antigens found on the surface of pathogens, such as membrane or tegumental proteins, have been analyzed and proposed as vaccine candidates due to their high probability of interaction with the host's immune system [45-48]."  Agree, but, it is not clear the relation of the ES proteins with this part of the text here.   Thank you for the observation. The text was modified as follows: “Further research to identify antigens from the ES proteins in the tegument of monogeneans could shed light on the development of better strategies to prevent or control fish diseases”.   11) Line 280: "Protein sequences were retrieved from an available R. viridisi transcriptome [62]." Mention a bit more information about these data, in particular the parasites, and early or late infection. That is important to understand the data here presented.   According to the suggestion, the following text was added in the revised manuscript: “…The specimens used for that transcriptome were all adults isolated from the gills of their fish host C. viridis, reared in laboratory conditions [62]. The predicted ES proteins were identified following the bioinformatic workflow of Gahoi et al. [19], which filter sequences based on signal peptide, discarding sequences with transmembrane regions, subcellular localization of mitochondrial proteins, endoplasmic reticulum retention signal, and GPI-anchor proteins.”  

Reviewer 4 Report

The paper describes the prediction and bioinformatic analysis of ES proteins from a monogenean fluke Rhabdosynochus viridisi. The study is interesting and shows that the analysis of the secretome of transcriptome basis is possible what is especially important in cases of limited access to pathogen specimens. However, as authors underline themselves, such results should be conformed using proteomic techniques. Not all transcripts are translated to proteins due to possible post-transcriptomic regulatory mechanisms.

I have a few comments:

Line 18: I suggest changing to: 1655 putative ES proteins were identified

Line 27: the statement “contribute to antigenicity” is  unclear, do Authors mean that ES proteins are/might be recognized as antigens by host immune response?

Figure 1 – “Percentage of genes” on the Y axis of the graph is not correct, as I understand the bars represent the percentage of all predicted ES proteins

Lines 108-109: this sentence is not clear, what do the 34 proteins found in the secretome relate to?

Line 282: In my opinion the bioinformatic workflow should be shortly described explaining on what basis the sequences encoding ES proteins were selected?

Author Response

We want to thank all reviewers for their constructive comments and suggestions. We have provided point-by-point answers to all the comments and hopefully we answered the questions. All modifications are highlighted in yellow in the main manuscript; all supplementary files were corrected; all Tables (except Table 4, now Table 1) became supplementary Tables; Figure 1 was corrected, and a graphical abstract was included.

Reviewer 4

1) Line 18: I suggest changing to: 1655 putative ES proteins were identified.   The line was changed according to the suggestion.   2) Line 27: the statement “contribute to antigenicity” is  unclear, do Authors mean that ES proteins are/might be recognized as antigens by host immune response?   The statement was corrected in the text: “…some may contribute to immunogenicity”.   3) Figure 1 – “Percentage of genes” on the Y axis of the graph is not correct, as I understand the bars represent the percentage of all predicted ES proteins.   Thank you fir the observation. The figure was modified as suggested.   4) Lines 108-109: this sentence is not clear, what do the 34 proteins found in the secretome relate to?   The sentence was corrected as follows: “Thirty-four ES proteins were annotated as cell adhesion molecules...”   5) Line 282: In my opinion the bioinformatic workflow should be shortly described explaining on what basis the sequences encoding ES proteins were selected?   Thank you fir the suggestion. The following text was added: “…which filter sequences based on signal peptide, discarding sequences with transmembrane regions, subcellular localization of mitochondrial proteins, endoplasmic reticulum retention signal, and GPI-anchor proteins”.

Round 2

Reviewer 2 Report

There have been relative few changes in the manuscript since the first submission and the basic flaws have not been substantially improved. The paper is still devoid of significance and interest. Why is characterizing potential E/S proteins in regards to protein domains, GO terms, KEGG pathways, and antigenicity predictions important or interesting?

It was previously suggested that the domains, GO terms, KEGG pathways etc of non-E/S proteins also be determined. But the authors summarily dismissed this without a good explanation. Doing the same analyses on non-E/S proteins could indicated an over (or under) representation of certain protein classes. For example, the authors state that 63% of the E/S protein are predicted to be antigens. If 63% of non-E/S proteins are also predicted to be antigens then you cannot conclude that E/S proteins are exceptionally antigenic. The same logic applies KEGG, GO, enzyme classes, etc. This information really needs to be included. The entire Results section is extremely boring text that really says nothing about E/S proteins. 

The authors indicate that the novelty of their work is that it is the first report of E/S proteins from the Diplectanidae family. However, it is never discussed how E/S proteins of the Diplectanidae are similar or different from the E/S proteins of other helminth families. A potentially useful study would be to do the same analyses from multiple groups of worms. 

Line 147, What specifically are the 'new insights'? It seems like the same speculations about E/S proteins. 

Line 305, the word immunomodulatory needs to be deleted. Immunomodulation was speculated. It was never demonstrated. The authors have only identifed potential E/S proteins with a computer program. 

The Conclusion section is primaily hypotheses. There were really no conclusions from the study. It was essentially a survey of a subset of genes. 

The sentence at lines 16-18 needs to be deleted. There was no mention of this in the body of the paper. 

Similarly, the two sentences at lines 23-27 could be deleted. Details of the hydrolases are not needed in the abstract.  

Author Response

We thank the reviewer for all comments and suggestions. Here we provide point-by-point answers. All changes in the manuscript are highlighted in yellow.

1) There have been relative few changes in the manuscript since the first submission and the basic flaws have not been substantially improved. The paper is still devoid of significance and interest. Why is characterizing potential E/S proteins in regards to protein domains, GO terms, KEGG pathways, and antigenicity predictions important or interesting?

The last paragraph of the Introduction was rephrased in order to highlight the significance of our work. Also, several parts of the discussion were modified to point out novel information for monogeneans in particular, or for platyhelminths in general. 

2) It was previously suggested that the domains, GO terms, KEGG pathways etc of non-E/S proteins also be determined. But the authors summarily dismissed this without a good explanation. Doing the same analyses on non-E/S proteins could indicated an over (or under) representation of certain protein classes. For example, the authors state that 63% of the E/S protein are predicted to be antigens. If 63% of non-E/S proteins are also predicted to be antigens then you cannot conclude that E/S proteins are exceptionally antigenic. The same logic applies KEGG, GO, enzyme classes, etc. This information really needs to be included. The entire Results section is extremely boring text that really says nothing about E/S proteins. 

We were not able to fully respond this point. We obtained the KEGG, GO and Pfam results for non-ES proteins of R. viridisi, which are presented in the Results section and supplementary files. However, further characterization (such as enzyme classes and antigenicity predictions) and other analyses would be difficult at this moment because of computational power limitations (there are 38171 non-ES proteins in the transcriptome and the server only analyzes 100 proteins at a time), so we would require more time than that provided by the journal to complete the revision. If the reviewer considers that this analysis is necessary in order to publish the article, we may need a couple of months.

3) The authors indicate that the novelty of their work is that it is the first report of E/S proteins from the Diplectanidae family. However, it is never discussed how E/S proteins of the Diplectanidae are similar or different from the E/S proteins of other helminth families. A potentially useful study would be to do the same analyses from multiple groups of worms. 

The discussion was reworded to specify comparisons with other helminths. 

4) Line 147, What specifically are the 'new insights'? It seems like the same speculations about E/S proteins. 

The sentence was rephrased.

5) Line 305, the word immunomodulatory needs to be deleted. Immunomodulation was speculated. It was never demonstrated. The authors have only identifed potential E/S proteins with a computer program. 

“Immunomodulatory” was deleted.

6) The Conclusion section is primaily hypotheses. There were really no conclusions from the study. It was essentially a survey of a subset of genes. 

The conclusion was partially modified.

7) The sentence at lines 16-18 needs to be deleted. There was no mention of this in the body of the paper. 

The sentence was deleted accordingly.

8) Similarly, the two sentences at lines 23-27 could be deleted. Details of the hydrolases are not needed in the abstract.  

The sentences were deleted as suggested.

Round 3

Reviewer 2 Report

Just 2 minor editorial comments: 

Line 127, the word 'only' is not needed

Lines 218-219, the word 'possibly' should be moved from the beginning of the sentence to between the words 'proteins' and 'have'.

Author Response

We thank the reviewer for these editorial comments.

Line 127, the word 'only' is not needed

The word “only” was deleted.

Lines 218-219, the word 'possibly' should be moved from the beginning of the sentence to between the words 'proteins' and 'have'.

The word possibly was moved as suggested by the reviewer.